

# Bee diversity in secondary forests and coffee plantations in a transition between foothills and highlands in the Guatemalan Pacific Coast

Gabriela Armas-Quiñonez[1,2,3], Ricardo Ayala-Barajas[4],
Carlos Avendaño-Mendoza[5], Roberto Lindig-Cisneros[2] and Ek del-Val[2]

[1] Posgrado en Ciencias Biológicas, Universidad Nacional Autónoma de México, Ciudad de México, Mexico
[2] Instituto de Investigaciones en Ecosistemas y Sustentabilidad, Universidad Nacional Autónoma de México, Morelia, Michoacán, Mexico
[3] Centro de Estudios Conservacionistas, Universidad de San Carlos de Guatemala, Guatemala, Guatemala
[4] Estación de Biología-Chamela, Instituto de Biología, Universidad Nacional Autónoma de México, Chamela, Jalisco, Mexico
[5] Escuela de Biología, Universidad de San Carlos de Guatemala, Guatemala, Guatemala

Corresponding author
Ek del-Val, ekdelval@iies.unam.mx

## ABSTRACT

**Background:** Although conservation of pristine habitats is recognized in many countries as crucial for maintaining pollinator diversity, the contribution of secondary forest conservation is poorly recognized in the Latin American context, such as in Guatemala. San Lucas Tolimán (SLT) is a high-quality coffee production region from the Atitlan Province, which has the second highest deciduous forest cover in Guatemala and pristine forest is prioritized for conservation. In contrast, secondary forest protection is undetermined, since these forests are normally removed or strongly affected by coffee farming practices. This situation may affect the diversity of native pollinators, mainly bees, which usually rely on the secondary forest for food resources.

**Methods:** We conducted a study to investigate the importance of secondary forests around the SLT coffee plantations (*Coffea arabica* L.) for pollinators. We compared bee diversity (richness, abundance and composition) in secondary forests of different age and coffee plantations with diverse farming techniques. Being the first study of pollinators in Guatemalan coffee plantations, we also recorded data for an entire year (2013–2014) in order to describe bee seasonality.

**Results:** We found significant differences in bee diversity between the coffee plantations and secondary forests, particularly early secondary forests showed higher bee abundances but diversity indices were similar between different vegetation type plots. In the early dry season, secondary forests showed the greatest native bee diversity. During the late dry season, when the coffee was flowering, honey bees were dominant in the same plots. This study provides important management insights to support the conservation of pollinators, since our results offer guidelines to improve coffee production by increasing native pollinator diversity.

Insect conservation, Guatemala, Insect diversity, Coffee

## INTRODUCTION

Rates of land-use change in primary forests are increasing worldwide, threatening biodiversity (*Montero-Castaño & Vilà, 2012*). As a result, secondary forests become alternative habitats and resource providers that promote a faunal diversity more characteristic of primary forest (*Peters et al., 2013*; *Taki et al., 2013*). In tropical forests, one of the most threatened groups of fauna interacting in both primary and secondary forests are the pollinators (*Winfree, Bartomeus & Cariveau, 2011*; *Cariveau & Winfree, 2015*), a fact that highlights the importance of also conserving secondary forests (*Taki et al., 2013*; *Winfree, Bartomeus & Cariveau, 2011*).

The conservation strategy in Guatemala for the past twenty-five years has been to preserve primary forest in situ (*National Congress of Guatemala, 1989*) by creating protected areas without management strategies to preserve the primary forest surroundings. As a result of this policy, the people in Guatemala are unaware of the role or ascribe little importance to secondary forest in terms of conserving biodiversity. In SLT in Sololá, Guatemala, secondary forest is also underestimated; here, the traditional and conventional coffee farmers focus their conservation efforts on the pristine forest, mainly to ensure ecosystem services such as pollination and water provision. This is a good strategy for the conservation of native pollinators who nest in this forest (*Jha & Dick, 2010*; *Klein, Dewenter & Tsccharntke, 2003*; *Klein et al., 2008*; *Rao & Stephen, 2010*; *Ricketts, 2004*; *Ricketts et al., 2008*). However, many native pollinators also require secondary forest to obtain food resources throughout the year (*Jha & Dick, 2010*; *Badano & Vergara, 2011*; *Klein et al., 2008*; *Kremen et al., 2004*) and these are also important for maintaining biodiversity in general (*Jules & Shahani, 2003*; *Kohler et al., 2008*; *Kremen et al., 2004*; *Kremen et al., 2007*; *Mandelik & Roll, 2009*).

As some authors have suggested (*Klein et al., 2007*; *Kremen, Williams & Thorp, 2002*; *Ollerton, Winfree & Tarrant, 2011*; *Winfree et al., 2007*), among the vertebrate and invertebrate pollinators, bees are the most important pollination agents. Bees are responsible for pollinating nearly two thirds of crops worldwide (*Brauman & Daily, 2008*; *Kremen, Williams & Thorp, 2002*); however, climate change and habitat fragmentation have endangered bee diversity and reduced bee populations, leading to a food crisis worldwide. Conservation of pollinators has therefore emerged as an issue of great importance (*Abrol et al., 2012*; *Ollerton, Winfree & Tarrant, 2011*).

Around the world, two species of coffee dominate the world market, *Coffea canephora* L. y *Coffea arabica* L., the second being the one that dominates more than 60% of it (*Ngo, Mojica & Packer, 2011*; *Enríquez et al. 2012*). The global coffee market has suffered price fluctuations, however this has not meant that the cultivation has decreased (*Ngo, Mojica & Packer, 2011*). In recent years, Guatemala has become the seventh largest coffee producer in the world and it is the most important crop in the country in terms of the employment and the foreign exchange that it produces (*Asociación Nacional del Café (ANACAFE), 2014*).

Sololá is a high-quality coffee region and crop fields are gaining territory at the expense of the forest due to the high demand for coffee produced in the region (*Asociación Nacional del Café (ANACAFE), 2014*; *Fischer & Victor, 2014*).

At present in SLT, the secondary vegetation commonly known as "monte" is normally cut down or treated with herbicides to prevent the secondary growth. Conventional farmers argue that by keeping the surroundings of the coffee clean they reduce the incidence of coffee pests, although there is no scientific evidence to support this belief. They also believe that by maintaining some primary forests they can guarantee pollinator diversity. On the other hand, the indigenous people who practice traditional farming are aware of the importance of secondary vegetation (G. Armas-Quiñonez, 2014, personal observation). Through their traditional knowledge, they know that these represent a habitat for numerous important species. However, they also cut down all the secondary growth in common areas, arguing that it is for the safety of the children and for aesthetic purposes. In both cases, the secondary vegetation is removed. As a consequence, the area with secondary vegetation in the region could be insufficient to maintain the community of native bee pollinators, especially when the coffee is not in flower. In other words, the traditional and conventional coffee farmers are not aware of the critical importance of secondary forest to the preservation of the pollinators, a situation that must be addressed in Guatemala since it could lead to loss of biodiversity and subsequently to deficient coffee production (*Philpott et al., 2008*; *Scheper et al., 2013*; *Steffan-Dewenter & Westphal, 2008*).

In order to ensure production and obtain other income sources, coffee farmers of the Guatemalan highlands have introduced *Apis mellifera* hives into their farms. It has been shown that coffee is mainly pollinated by the honey bee but also frequently visited by stingless bees (*Ngo, Mojica & Packer, 2011*). However, the raising honey bees within coffee farms is practice may present some risk since it could have unknown impacts on native bee populations (*Badano & Vergara, 2011*; *Garibaldi et al., 2011*; *Shavit, Dafni & Ne'eman, 2009*; *Van Engelsdorp & Meixner, 2010*; *Winfree et al., 2007*), particularly in Guatemala where such interactions are poorly studied.

Coffee farming in Guatemala is very heterogeneous in terms of farming techniques. Big farms have changed from traditional management to conventional and highly intensive farming techniques that usually include higher inputs of agrochemicals, mainly pesticides, or on rare occasions have changed to integrated pest management. At the same time, traditional farmers use multi-farming techniques, where small coffee plantations are cultivated using intercropping with several banana hybrids (*Musa × paradisiaca*), papaya (*Carica papaya*), macuy (*Solanum americanum*), besides other crops. Traditional farmers usually cannot afford agrochemicals (pesticides, herbicides and fertilizers) to spray on their fields, and this management could therefore be contributing more than conventional coffee cultivation to the maintenance of bee diversity (*Schmitt, 2006*; *Schüepp, Rittiner & Entling, 2012*). However, in Guatemala, there have been no studies published that address this issue.

This study was therefore conducted in order to investigate the importance of secondary forests in maintaining pollinator diversity around the SLT coffee fields. To accomplish this

 

**Table 1  Description of plots for the three studied sites.**

| Coffee plantation farms | Type of plots | Coffee farming techniques | Adjacent forest management |
|---|---|---|---|
| 1 | Early: short height secondary forest<br>Late: medium height secondary forest<br>Coffee: traditional coffee field, shaded with native species | Traditional farming: traditional methods of pest removal, rare use of chemical fertilizers and pesticides. | Community forest management. The community regulates and controls the use of the forest, creating a preserved forest. |
| 2 | Early: short height secondary forest<br>Late: medium height secondary forest<br>Coffee: shaded coffee with *Grevillea robusta* and *Inga* sp. | Conventional with high intensity farming practices: controlled production, integrated pest control with minimum use of pesticides and herbicides. | Private reserve. Forest with low human disturbance and no access granted to the local people, creating in a preserved forest. |
| 3 | Early: short height secondary forest<br>Late: medium height secondary forest<br>Coffee: shaded coffee with *Grevillea robusta*. | Conventional with low intensity farming practices: uncontrolled production and pest control, occasional use of pesticides and herbicides. | Private reserve. Forest with human intervention and low control of access for local people, producing a disturbed forest. |

goal, the study was designed to compare bee diversity in plots featuring different stages of secondary forest and in coffee fields managed under a range of farming practices.

# MATERIALS AND METHODS

## Study sites

This study was conducted from March 2013 to February 2014 in SLT foothills region (Appendix 1A). SLT is located at the limit between the central highlands and costal lowlands in southeastern Guatemala, where many Kaqchikel indigenous people live. SLT is bordered by two volcanoes, Atitlán and Tolimán that range from 800 to 3,500 masl and produce a variety of microhabitats. According to the Villar classification, the primary vegetation of studied farms are within a subtropical humid forest, where broadleaf evergreen forest divides the mountain forest from the tropical humid savanna on the Pacific coast (*Villar Anleu, 1998*). These biotic and topographic differences give the area a dynamic ecotone with high precipitation (*Villar Anleu, 1998*, *2003*). According to the Guatemalan National Council of Protected Areas (*Consejo Nacional de Áreas Protegidas (CONAP), 2008*), the department of Sololá, where SLT is located, has 35% forest cover, making it the most forest-covered department in Guatemala. This fact is associated with the high degree of community conservation but is also due to the presence of private farms that normally have their own forest reserves.

## Sampling design

Sampling was conducted at three coffee farms with different management types (Table 1). The farms harvest *Coffea arabica*, cultivar "caturra". The three farms have secondary forests nearby, farm 1 and 3 are separated by 0.8 km, and 4 km from farm 2 (Appendix 1B). The names of these private farms must be withheld at the request of the owners.

In each of the studied farms, three plots of 60 m$^2$ were established. Each plot was categorized as early secondary growth, late secondary growth or coffee plantation, the plots within the farm were separated by at least 0.5 km depending on the farm. Early secondary growth was characterized by early secondary forest with up to one year of development,

mainly herbaceous vegetation with few bushes and an abundant incidence of light. Late secondary growth was characterized by late secondary forest with two to three years of succession, with mainly shrub vegetation and luminosity slightly restricted below the high bushes. Coffee plots were selected in patches of shaded coffee in the selected farms. Plot characteristics are presented in Table 1.

## Bee sampling

Bees were sampled every month from March 2013 to February 2014 in each vegetation type plot, covering the two Guatemalan climatic seasons established as dry or summer from November to April and rainy or winter from May to October (*Consejo Nacional de Áreas Protegidas (CONAP), 2008*). During three days each month, five people searched for bees on flowering plants in all of the selected plots. For each flowering plant species in each plot, bee sampling was conducted for 40 min at different times between 8:00 and 12:00 p.m. The sampling schedule was done considering results of previous temperature and humidity monitoring where optimal conditions for bee activity in the area were established. Bee sampling was based on the direct search method on flowers, using net sweeping (*Brosi et al., 2008*; *McGavin, 1997*; *McMullen, 1965*). Bees captured from flowers were killed by freezing in individual containers. Native bee (no honey bee) specimens were mounted on insect pins labeled with field data and assigning a unique code for deposition in the "Colección de abejas nativas del Centro de Estudios Conservacionistas de la Universidad de San Carlos de Guatemala". In contrast, honey bee specimens were sampled and recorded but not mounted. Taxonomical keys were used for bee identification to genus or species (wherever possible) (*Ayala, 1990*, *1999*; *Michener, 2007*; *Michener, McGinley & Danforth, 1994*; *McGinley, 1986*; *Roberts, 1972*; *Roubik & Hanson, 2004*; *Smith-Pardo, 2005*; *Snelling, 1974*) and with the help of the bee expert Ricardo Ayala, and collaboration of Mabel Vásquez, Carmen Yurrita y María José Dardón. Collection 166 permits for conducting bee and plant sampling are 000960, 002011, 002009 and the Field 167 Research License were 007/2015 and 043/2012.

## Data analysis

### Seasonal bee and plant abundance across vegetation types

In order to analyze changes in plant and bee abundance per season and vegetation type, we used nested ANOVAs considering several response variables: plant richness, total bee abundance, stingless bee abundance, social bee abundance, honey bee abundance and native bee abundance, as well as abundance per family and we included vegetation type, season and the interaction between vegetation type and season as explanatory variables, we also added an error term to account for spatial and temporal pseudoreplication considering farm/season/vegetation type. All response variables were log transformed to comply with ANOVA requirements. Also, Pearson correlations were performed in order to determine whether bee richness was correlated with plant richness, and also if honey bee incidence, was correlated with other native bee groups over the study seasons. We grouped the bees into honey bees, native bees and stingless bees with the aim of differentiating the contribution of introduced honey bees, from stingless bees, which are

the second largest group of bees that visit coffee (*Ngo, Mojica & Packer, 2011*) and also of the rest of native bees that are generally seasonal.

### Bee diversity in coffee fields and the surrounding secondary forest

Global bee diversity (considering all the sampling period) was evaluated by comparing the secondary forests and coffee plantations, with the three different farming management type (Table 1). For richness, we used the Chao1 index and Abundance-based Coverage Estimator (ACE), in order to take both rare and abundant bee species into account. The incidence of both rare and common bee species was estimated with the Chao2 and Incidence Covered Estimator (ICE) indices. Diversity was calculated by Shannon-Wiener (H). These indices were calculated using the EstimateS Program (*Colwell, 2013*). All the diversity indices were analyzed using ANOVAs, taking into account the vegetation type and farming management type as explanatory variables.

Finally, a cluster analysis with Euclidean distances with 100 bootstrap samples was performed, using the bee diversity data to look for similarities between the vegetation and farming techniques (farms) and to infer the importance of the coffee management for the bees. All the analysis were performed using the R program (*R Core Team, 2014*) using the packages *stats, vegan* (*Oksanen et al., 2019*) and *Pvclust* (*Suzuki, Terada & Shimodaira, 2019*).

## RESULTS

### Seasonal richness and abundance

Over one year of study, 3,004 bee specimens, belonging to 102 species (Appendix 2) and 100 species of flowering plants with visiting bees (Appendix 3) were recorded. Bee and plant richness inside plots were significantly correlated ($t_{36} = 7.82$, $p = 2.8E{-}09$, cor = 0.79) showing a close relationship between them (Fig. 1A).

The study area showed a sparse flowering season from March to October 2013 and high flowering in the early dry season from November 2013 to February 2014, closely reflecting the two climatic seasons in Guatemala. A few months after the lowest records of bee abundance (March–October 2013), the rainy season promoted vegetative growth in the secondary forest, and it was correlated with the highest values of flowering plant richness with a 21% increase (Fig. 2; Appendix 3).

Plant richness was similar between seasons ($F_{1,\ 1} = 2.53$, $p = 0.37$), as well as total bee abundance and richness ($F_{1,\ 1} = 40.6$, $p = 0.09$ and $F_{(1,\ 1)} = 66.1$, $p = 0.07$, respectively). However, bee groups responded in different ways, native bees and social bees showed higher abundances during the dry season ($F_{(1,\ 1)} = 195.4$, $p = 0.05$, $F_{(1,\ 1)} = 118.5$, $p = 0.05$, respectively) while honey bee and stingless bee abundance showed a similar seasonal trend but was no statistically significant ($F_{1,\ 1} = 19.1$, $p = 0.1$; $F_{1,\ 1} = 72.9$, $p = 0.07$, respectivelysee Table 2). There was an interesting finding in February 2014 (Fig. 2) when bee richness increased by 57% (from 28 to 53 species), presenting the highest value in the study, but out of phase with respect to the peak of local flowering plants (December 2013), coinciding with the beginning of drought, with the biggest drop of flowering (14%). In the following month when the coffee plants were flowering

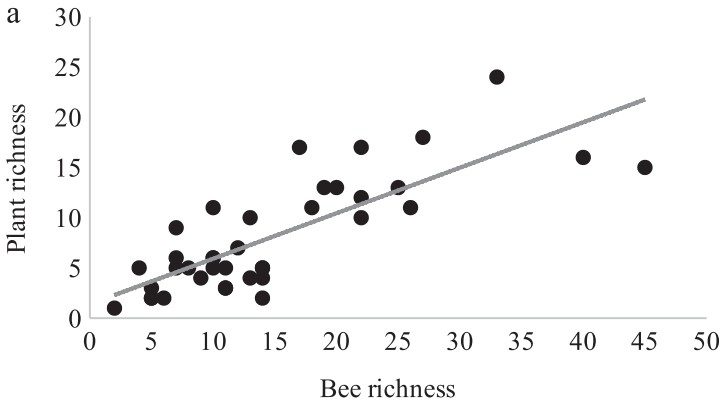

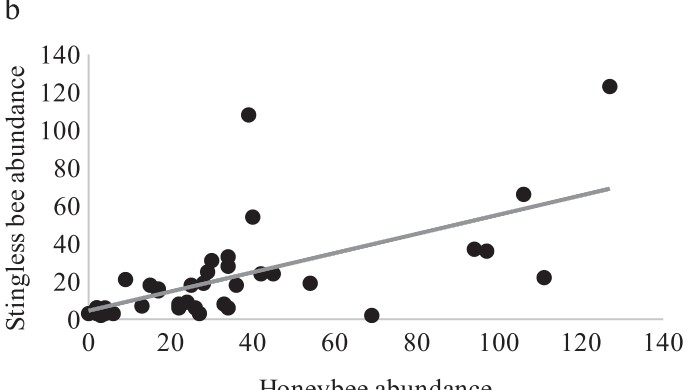

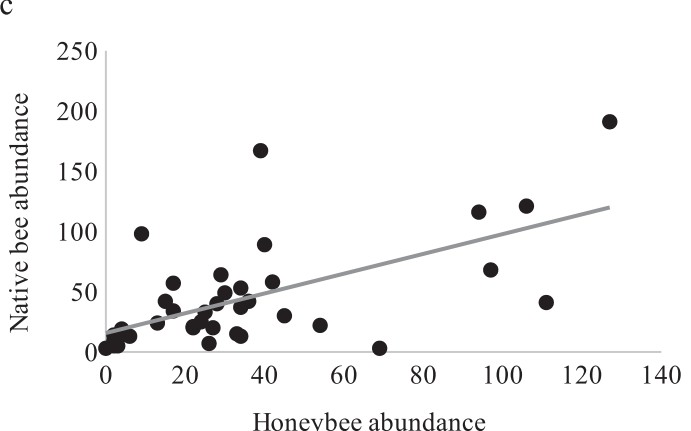

**Figure 1 Bee richness and plant richness.** Pearson correlations between: (A) Bee richness and plant richness ($r = 0.79$, $p = 2.8E-09$). (B) Honeybee and stingless bee abundance ($r = 0.62$, $p = 2.95E-05$). (C) Honeybee and native bee abundance ($r = 0.61$, $p = 4.84E-05$).

(March 2014), the greatest number of bees was recorded (437 records), but the native bee species began to decrease (10%).

Interestingly, honey bees and stingless bees were significantly and positively correlated ($r = 0.62$, $t_{36} = 4.78$, $p = 2.95E-05$, Fig. 1B), as were the honey bees and native bees ($r = 0.61$, $t_{36} = 4.61$, $p = 4.84E-05$, Fig. 1C). Highlighting that for the three groups of bees, the availability of floral resources is important for their activity.

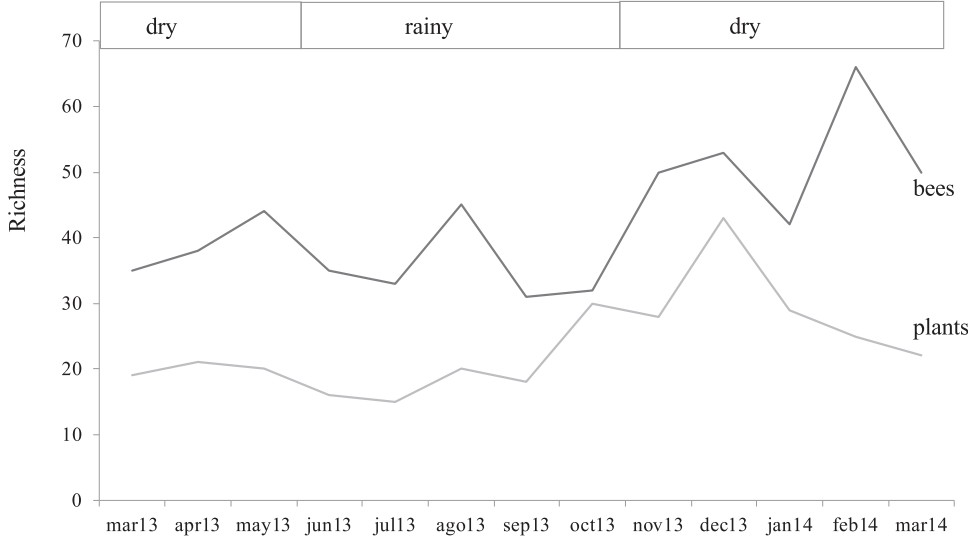

**Figure 2 Bee and flower through time.** Variation of bee and flowering plant richness in months in which the samplings were conducted highlighting the two climatic seasons for the region.

**Table 2 Nested Anova results with significant *p* values shown in bold numbers.**

| Variables | df | Honey bees | | Native bees | | Stingless bees | | Social bees | | Total bee abundance | | Total bee richness | | Plant richness | | Andrenidae | | Apidae | | Colletidae | | Halictidae | | Megachilidae | |
|---|---|---|---|---|---|---|---|---|---|---|---|---|---|---|---|---|---|---|---|---|---|---|---|---|---|
| | | *F* | *p* | *F* | *p* | *F* | *p* | *F* | *p* | *F* | *p* | *F* | *p* | *F* | *p* | *F* | *p* | *F* | *p* | *F* | *p* | *F* | *p* | *F* | *p* |
| Season | 1, 1 | 19.1 | 0.14 | 195.4 | **0.05** | 72.9 | 0.07 | 118.5 | **0.05** | 40.6 | 0.09 | 66.1 | 0.07 | 2.53 | 0.37 | 1.69 | 0.41 | 39998 | **0.01** | 27.82 | 0.11 | 1.33 | 0.45 | 0.71 | 0.75 |
| Vegetation type | 2, 8 | 6.2 | **0.02** | 5.33 | **0.03** | 8.41 | **0.01** | 9.01 | **0.008** | 7.92 | **0.01** | 6.38 | **0.02** | 6.95 | **0.02** | 0.82 | 0.47 | 10.01 | **0.007** | 0.84 | 0.46 | 2.43 | 0.15 | 19.01 | **0.0009** |
| Season: vegetation type | 4, 8 | 2.8 | 0.12 | 0.27 | 0.77 | 0.36 | 0.71 | 2.39 | 0.15 | 1.58 | 0.26 | 0.71 | 0.51 | 0.31 | 0.74 | 0.06 | 0.93 | 0.59 | 0.57 | 0.29 | 0.75 | 0.01 | 0.98 | 3.59 | 0.08 |

## Bees in coffee plantations and the surrounding secondary forest

The farms presented different bee richness and abundance. As shown in Fig. 3, farm 1 had the highest number of bees followed by farm 2 and farm 3. Farms 1 and 2 presented the highest bee abundance in plots with secondary forest, while in farm 3 the coffee plantations presented the highest abundance.

Bee richness mainly comprises five families (Fig. 3). Most of the bees are Apidae (84.8%), followed by Halictidae (8.9%), Andrenidae (2.6%), Megachilidae (2.3%) and Colletidae (1.5%). Species from the five bee families were registered in the three farms. On farm 3, 91% of captures belonged to Apidae, while farms 1 and 2 had a higher representation of Colletidae and Andrenidae. Vegetation type (early secondary forests, late secondary forests and coffee plantations) showed a significant effect on total bee abundance ($F_{2, 8} = 7.92$, $p = 0.01$), bee richness ($F_{2, 8} = 6.38$, $p = 0.02$), native bees ($F_{2, 8} = 5.33$, $p = 0.03$), stingless bees ($F_{2, 8} = 8.41$, $p = 0.01$), social bees ($F_{2, 8} = 9.01$, $p = 0.008$), honey bees ($F_{2, 8} = 6.2$, $p = 0.02$), and Apidae and Megachilidae families ($F_{2, 8} = 10.01$, $p = 0.007$, $F_{2, 8} = 19.01$, $p = 0.0009$, respectively), particularly between early growth and coffee plantation (Fig. 4).

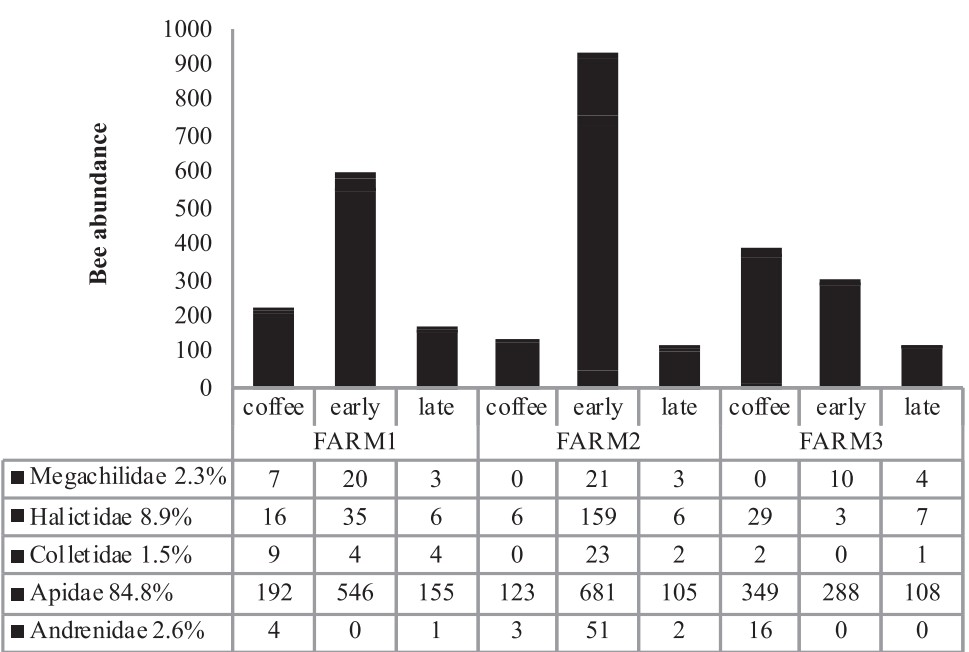

| | coffee | early | late | coffee | early | late | coffee | early | late |
|---|---|---|---|---|---|---|---|---|---|
| | | FARM1 | | | FARM2 | | | FARM3 | |
| ■ Megachilidae 2.3% | 7 | 20 | 3 | 0 | 21 | 3 | 0 | 10 | 4 |
| ■ Halictidae 8.9% | 16 | 35 | 6 | 6 | 159 | 6 | 29 | 3 | 7 |
| ■ Colletidae 1.5% | 9 | 4 | 4 | 0 | 23 | 2 | 2 | 0 | 1 |
| ■ Apidae 84.8% | 192 | 546 | 155 | 123 | 681 | 105 | 349 | 288 | 108 |
| ■ Andrenidae 2.6% | 4 | 0 | 1 | 3 | 51 | 2 | 16 | 0 | 0 |

**Figure 3 Total bee abundance per farm and vegetation type plots.** Cumulative abundance records per farm and plot are shown in the upper part of the figure. The lower table shows the total abundance of records per bee family.

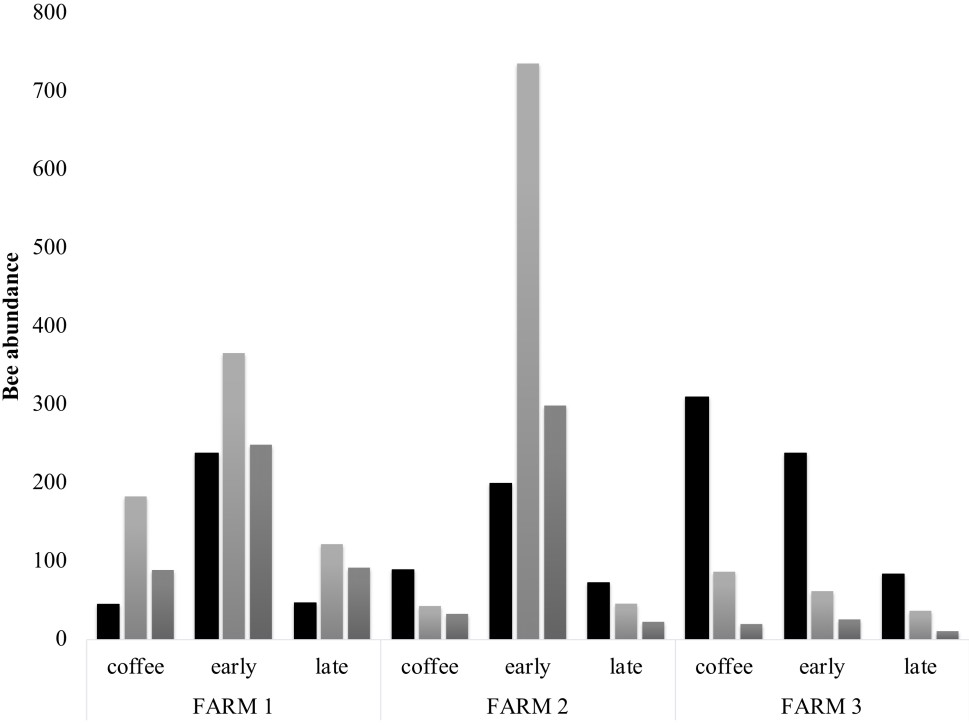

**Figure 4 Bee diversity per farm.** Bee abundance per farm and vegetation type plots. Honeybee abundance is shown in black, native bee abundance in light grey and stingless bee abundance in dark grey.
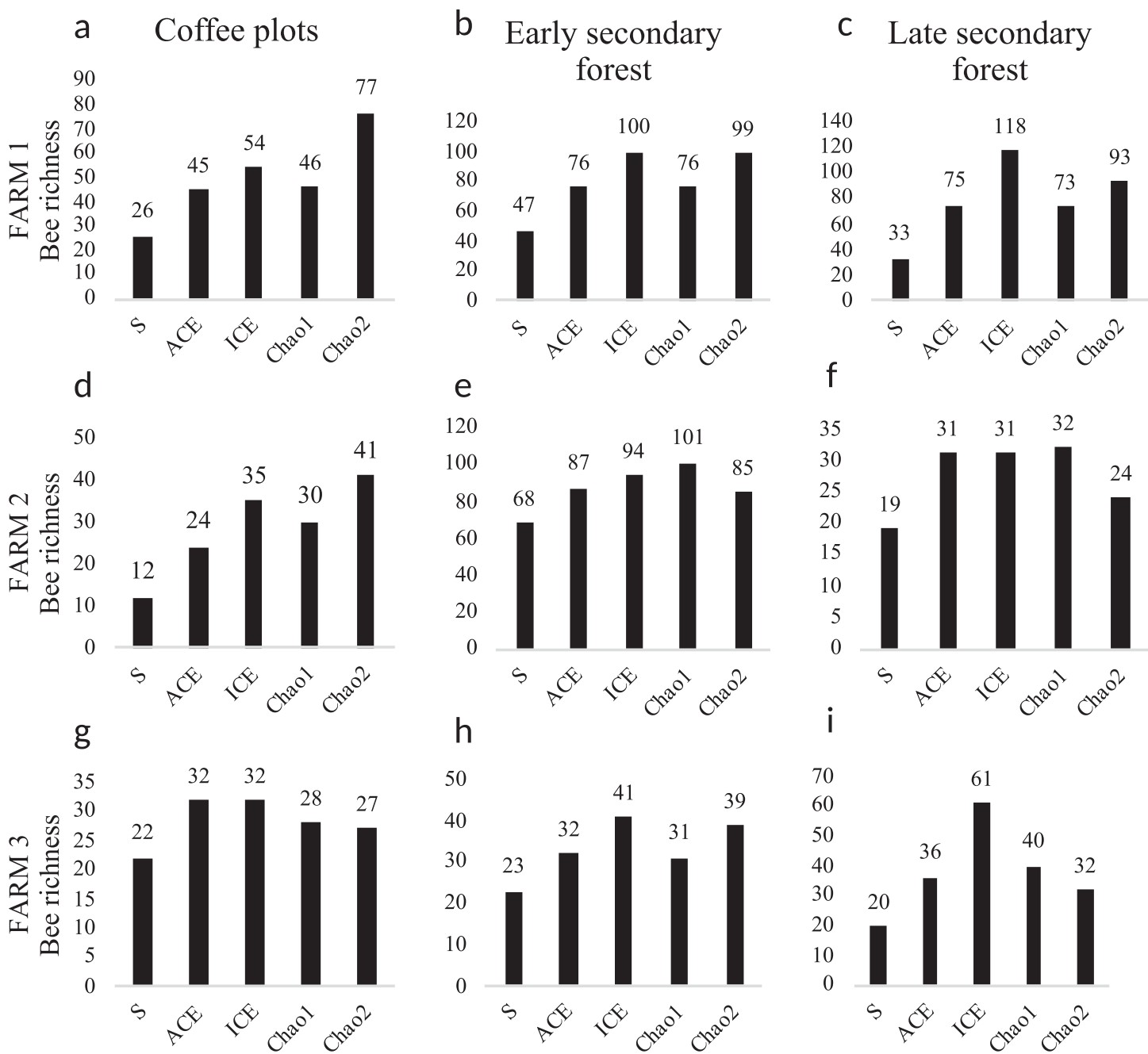

**Figure 5 Bee diversity per farm and vegetation type.** Bee species diversity per farm and vegetation type plots. (S) Bee richness. (ACE) Abundance-base coverage richness estimator -$S_{ACE-}$. (ICE) Incidence coverage estimator -$S_{ICE}$-. (Chao1) Richness estimator -$S_{Chao1}$-. (Chao2) Incidence estimator -$S_{Chao2}$-. Coffee plots are shown in (A), (D) and (G); Early secondary forests in (B), (E) and (H) and Late secondary forests in (C), (F) and (I).    

The diversity estimators Chao1, ACE, ICE (Fig. 5) calculated per farm and vegetation type did not present any significant difference ($p > 0.05$). The Chao2 estimator showed significant differences among farms ($F_{2, 2} = 7.903$, $p = 0.048$). The Shannon-Wiener diversity index (H) did not present significant differences among farms. In farms 1 and 2,

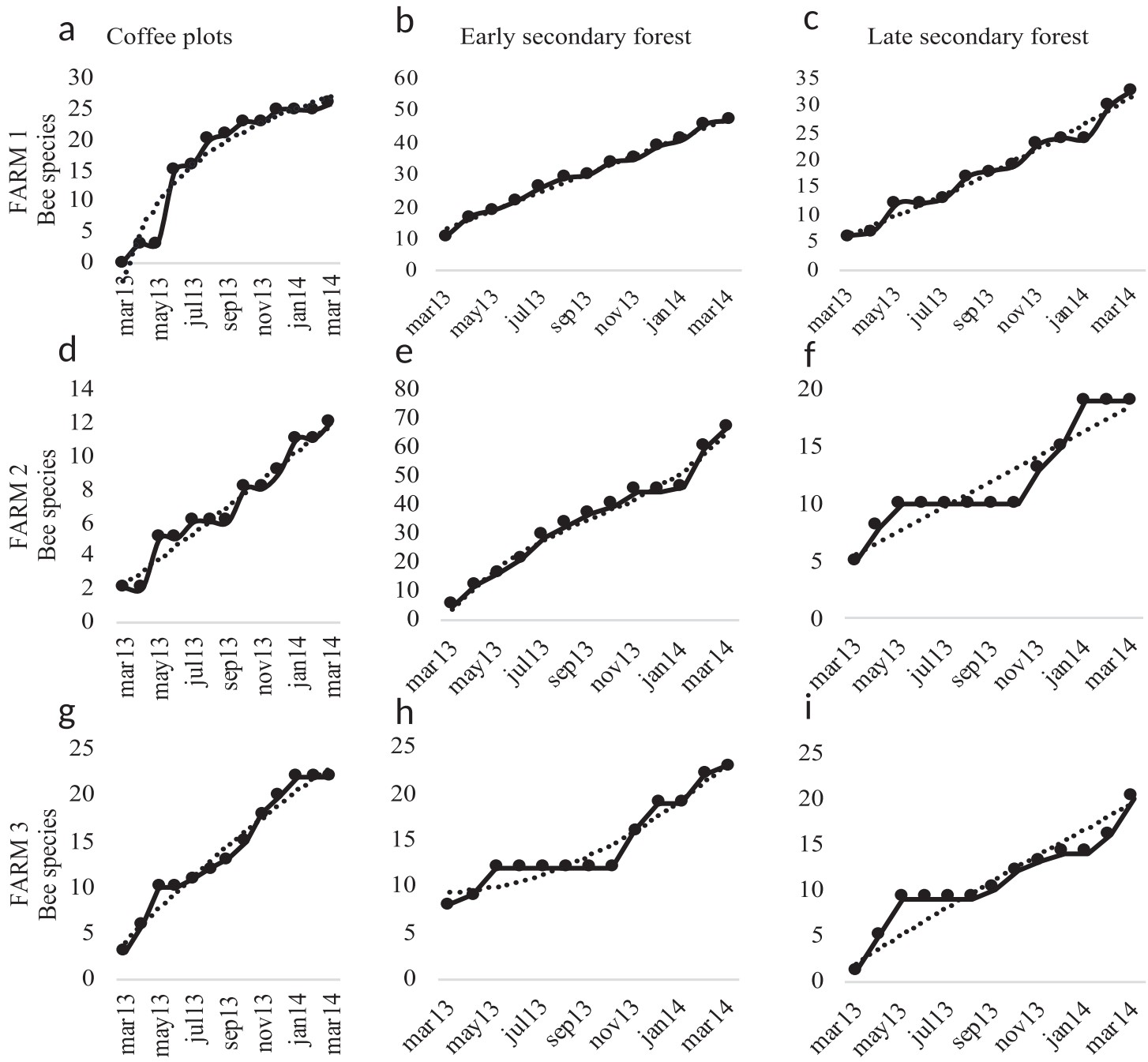

**Figure 6 Species accumulation curves.** Species accumulation curves of bee richness obtained in the study per farm and vegetation type plots. Coffee plots are shown in (A), (D) and (G), Early secondary forests in (B), (E) and (H) and Late secondary forests in (C), (F) and (I).

the richness estimators gave higher values to plots with early secondary forest, while the highest estimate value on farm 3 was for the coffee plantation. The species accumulation curves showed stabilized curves that started to flatten down in some plots (Fig. 6). Coffee plantations registered the lowest values in bee diversity, and accordingly, the richness estimators predicted the lowest number of species (Figs. 5 and 7).

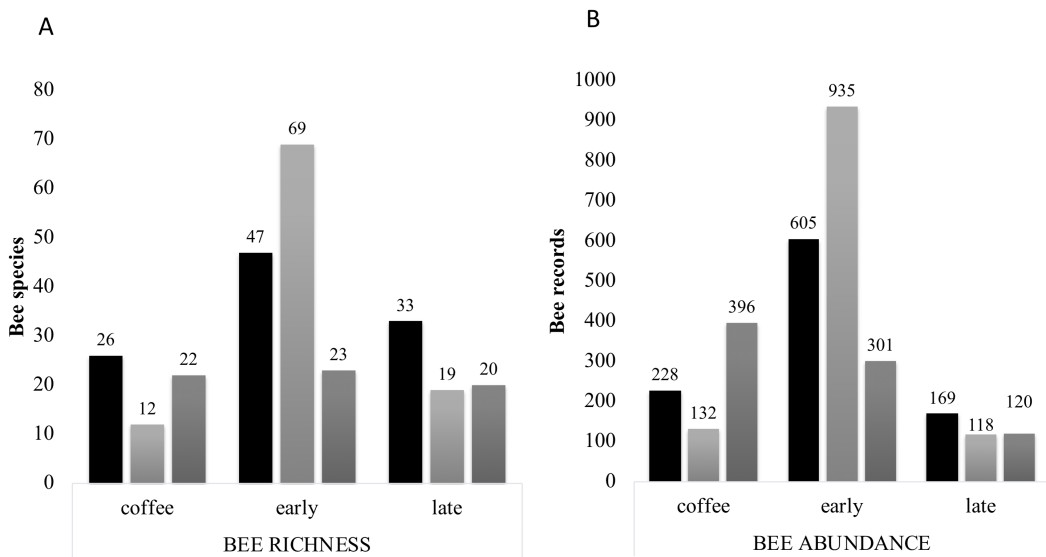

**Figure 7 Bee richness per far and plot type.** (A) Total bee richness and (B) abundance registered per vegetation type plots. Black bars show farm 1 data, light grey bars show farm 2 data and dark grey bars show farm 3 data. The total richness and abundance values are shown in the numbers above the bars.

In farms 1 and 2, the coffee plantations registered most of the bee records during the coffee flowering season (March 2014), otherwise, neither bee activity nor early growth vegetation were recorded because coffee farmers clean adventive vegetation from coffee plantations, particularly in farm 2. On farm 3, the coffee was rarely cleaned in this manner, which contributed to the establishment of early secondary forest vegetation.

An important finding was the presence of a new species of bee genus *Rhathymus* Lepeletier & Serville, 1828, *Rhathymus atitlanicus*, described in *Ayala, Hinojosa-Díaz & Armas-Quiñónez (2019)*. Those bees were only found in secondary forest of farm 1 and farm 3 (Appendix 2).

Finally, the cluster analysis of bee diversity per vegetation type (Fig. 8) presents a significant pattern that groups the study plots into two significant aggregations. The strongest aggregation with 97% confidence is composed by the early secondary forests from farm 1 and farm 3 and the coffee plots of farm 3 (the farm that allows secondary forest plants into the coffee plantation). The other aggregation with 97% confidence, is composed by the coffee and late secondary forests of all three farms. Within this group there are significant differences among the plots of farm 1, where coffee is harvested following traditional practices, and on the other side of the same group, is the late secondary forest linked with the coffee of farm 2. Cluster analysis shows that the early secondary forest in farm 2 is 77% different to the rest of the plots.

# DISCUSSION
## The effect of flowering seasonality on bees
The bees recorded over the whole year in secondary forest and coffee fields could give an insight into the synchronization that exists between bees and phenology of flowering

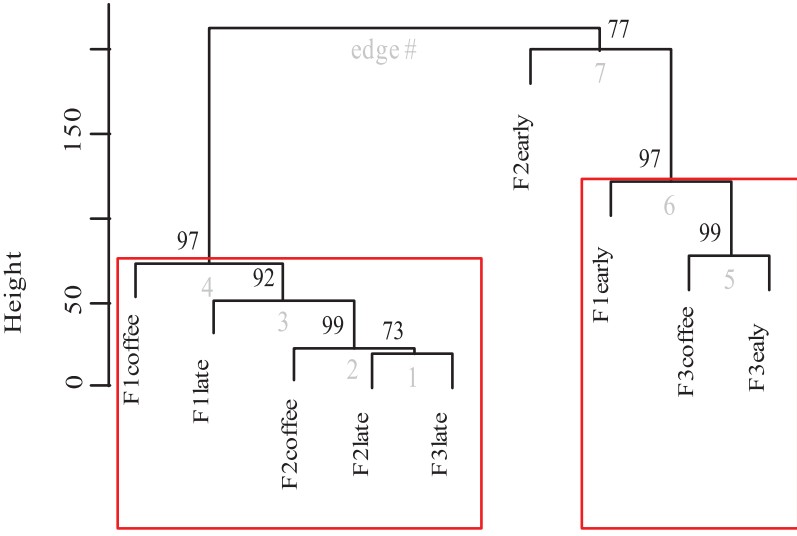

**Figure 8 Dendrogram of bee species composition per farm and plot type.** Cluster dendrogram of bee species abundances recorded between plots of the three farms (cluster analysis calculated in R using Vegan and Pvclust packages and Euclidean distance). The *p*-value is shown at the top of each edge, presented as a percentage value of confidence, in which a value of 95 or higher represents a significant supported data aggregation. The two significantly different groups are indicated by red rectangles.

plants, due to the periods of highest bee abundance matching with flowering periods and the correlation between these parameters throughout the study. This correlation supports previous findings (*Brosi, 2009*; *Banks et al., 2014*; *De Marco, Monteiro & Coelho, 2004*; *Donald, 2004*; *Taki et al., 2013*) about the great importance of secondary forests in the provision of resources to bees. This seasonality can have some important consequences for bee diversity but also for coffee production, since the highest abundances of native bees were observed during secondary forest flowering (March 2014), close to the coffee flowering that varies according to the first rains. Early flowering of coffee could coincide with secondary forest flowering, which would cause native bees to interact with the high honey bee abundance possibly giving rise to a saturated environment of pollinators for the coffee and secondary forest. However, the asynchrony of pollinators and flowering periods in coffee fields must be treated with care and taken into account in crop management as a priority for improved of coffee production (*Boreux et al., 2013*).

When investigating the relationship between honey bee and native bee abundance, we found a positive correlation between them, as other studies in Mexican coffee plantations have found (*Badano & Vergara, 2011*). This fact suggests that honey bees may not yet have saturated this ecosystem, and that, according to *Banks et al. (2013)*, the contribution of pollinators from nearby forests to the coffee plantations is still high. This notion can be supported by the fact that honey bees only maintain high populations during coffee flowering, otherwise, their cultured populations are kept to a minimum in the area. However, the abundance of the groups bees and the total of bees are showing differences between vegetation types probably due to the nearby farm vegetation,

especially surrounding secondary vegetation and primary forest were the native bees usually nest.

## Secondary forests as coffee pollinator enhancers

The data of bee abundance, richness and diversity supports the importance of secondary forests for bee pollinators (*Arnan et al., 2011*; *Carvalheiro et al., 2012*; *Banks et al., 2013*; *Brosi et al., 2008*; *Boreux et al., 2013*), not only to maintain bee diversity, but also to improve coffee production in the Pacific Coast Foothills of Guatemala. The significant results in the nested anovas remarks how the vegetation interact with bees, they are most abundant in early secondary forests(Table 2). Meanwhile, the presence of stingless bees in the secondary forests and also in the flowering coffee plants, suggests that these bees, which depend exclusively on the forest for nesting, also depend on secondary forests for the acquisition of essential resources (*Winfree, 2010*).

Secondary forests showed that they can maintain and provide resources for the native bees during periods when the coffee is not flowering. The farmers that use conventional agriculture need to place greater emphasis on preserving secondary forests, rather than only pristine forests (*Blanque, Ludwing & Cunningham, 2006*). Pollination tests are required in order to compare the pollination efficiency of native bees and honey bees in these plantations. In this way, it would be possible to more definitively establish the importance of preserving native bees and the places where they obtain resources, such as the secondary forest, for coffee production (*Winfree, 2010*).

## Effect of farm management

We found important differences in bee diversity between farms. Farm 1 shows the highest diversity values, suggesting that, despite the lack of high technology, traditional knowledge remains effective in preserving native bee diversity, especially for the stingless bees. The conventional farm with high-intensive management (farm 2) keeps the coffee plantations cleared of early secondary forest but maintains surrounding secondary vegetation on the farm; therefore, the presence of *Tephrosia* spp. could function as a provision plant for native bee species. A positive interaction between farm 2 and bee family abundance can suggest the importance of nearby forest, since the farm 2 manage their own forest reserve, providing the bees with a complex landscape that may enhance bee resource acquisition, as suggested previously by different authors (*Carrié et al., 2017*; *Winfree et al., 2009*).

Also, it is evident that the conventional farming with low-intensity management and a disturbed surrounding forest (farm 3) has the lowest bee diversity and, that it is the early secondary forest left in the coffee fields for a short time period that functions as resource provider for the surrounding bee diversity. This could explain why, in the cluster analysis (Fig. 8), the coffee of farm 3 was grouped together with the early secondary forest of farm 1 and 2.

None of the studied farms use honey bee breeding to pollinate coffee at the time of this study. Some neighboring farms, however, do have managed bee hives and it is possible that the honey bees recorded during the study came from those neighboring farms.

The farm with the closest neighboring honey bee hives was farm 3, which showed the lowest bee abundance, richness and diversity values as well as the highest honey bee abundance. On the other hand, this farm (3) also shows a poor surrounding forest management providing few resources for native bees, contrary of farm 1 that promotes this mixed-landscape (*Carrié et al., 2017*).

Another factor to consider that may affect bee diversity is the use of chemicals in the farms: the quantity of insecticides used for pest control inside (such as that used to control the Mediterranean fruit fly), but also those used outside farms in neighboring crops cultivated near coffee (*Brittain et al., 2010*; *Schmitt et al., 2009*; *Schüepp, Rittiner & Entling, 2012*).

## CONCLUSIONS

In coffee plantations the presence of secondary forest in the early growth stage, shows a significant positive effect on bees, and taking into account the importance of bees for pollination this is a natural way to increase the pollinators. Regarding farm management, this study can be used to apply certain strategies that would benefit native bee diversity around the country. Through incorporating some traditional farming techniques into conventional coffee field management, such as letting the surrounding secondary forests grow at least in the early dry season, it can be demonstrated that these provide floral resources for native bees that will also visit the coffee during its flowering stage. Our results also support the value of traditional farming, which in this study demonstrated a high diversity of native bees, capable of pollinating small coffee plots and thus saving the cost of maintaining honey bee hives for pollinating small crops. However our sampling effort was restricted to three farms in a particular region of Guatemala, therefore in order to be able to generalize our findings we should increase our sampling effort.

## ACKNOWLEDGEMENTS

We gratefully acknowledge the "Posgrado en Ciencias Biológicas, UNAM". This publication of Gabriela Armas is part of the doctoral thesis in the "Programa de Doctorado en Ciencias Biológicas, Universidad Nacional Autónoma de México (UNAM)". We also acknowledge the farms personnel for their help with the field work and the farm owners and communitarians who allowed the entrance and the data record on their properties. We sincerely acknowledge the "Laboratorio de Entomología Aplicada y Parasitología", "Programa de Experiencias Docentes con la Comunidad" and "Centro de Estudios Conservacionistas", Universidad de San Carlos de Guatemala, for having provided materials and equipment during the collection, processing and data analysis.

### Funding

This work was funded by the National University of Mexico. The funders had no role in study design, data collection and analysis, decision to publish, or preparation of the manuscript.

## Grant Disclosures

The following grant information was disclosed by the authors:
National University of Mexico.

## Competing Interests

The authors declare that they have no competing interests.

## Author Contributions

- Gabriela Armas-Quiñonez conceived and designed the experiments, performed the experiments, analyzed the data, prepared figures and/or tables, authored or reviewed drafts of the paper, and approved the final draft.
- Ricardo Ayala-Barajas conceived and designed the experiments, authored or reviewed drafts of the paper, and approved the final draft.
- Carlos Avendaño-Mendoza conceived and designed the experiments, analyzed the data, authored or reviewed drafts of the paper, and approved the final draft.
- Roberto Lindig-Cisneros conceived and designed the experiments, analyzed the data, authored or reviewed drafts of the paper, and approved the final draft.
- Ek del-Val conceived and designed the experiments, performed the experiments, analyzed the data, authored or reviewed drafts of the paper, and approved the final draft.

## Field Study Permissions

The following information was supplied relating to field study approvals (i.e., approving body and any reference numbers):

Bee and plant collections were approved by the Consejo Nacional de Áreas Naturales Protegidas de Guatemala with the following permits: 000960, 002011, 002009 and Licencia de Investigación 007/2015 and 043/2012.

## Data Availability

The raw measurements are available in the Supplemental Files.

## Supplemental Information

Supplemental information for this article can be found online at http://dx.doi.org/10.7717/peerj.9257#supplemental-information.

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
