# Peer review of "Bee diversity in secondary forests and coffee plantations in a transition between foothills and highlands in the Guatemalan Pacific Coast"

_PeerJ, doi:10.7717/peerj.9257_

## Round 0.1 · original submission · Minor Revisions

Please address the comments from the two reviewers. I look forward to receiving the edited manuscript.

Reviewer 1 ·

Basic reporting

The paper is well-written. The study is well-justified in terms of knowledge gaps and the application of the results. The introduction and discussion draw from the literature to an appropriate degree. Figures and tables seem clear.

Experimental design

There seems to be a potential weakness in the design in terms of statistical analysis. There are three farms, each with three subplots of different land type in each farm. It is not clear how far apart these farms are or how distant the subplots within a farm would be. As such, there are only three replicates per treatment (i.e farm), so it makes it challenging for stats. There was quite a bit of sampling effort per farm, but it's repeated measures so the non-independence of the samples should be accounted for in the analyses. It's not clear to me if, how or when that happened. There is some reference to linear mixed-effect models, but I don't see any indication of the model structure (i.e. fixed vs random effects). ANOVAs are used throughout, but I wonder if that is the best approach. I would expect the error structure of bee collection data to be more appropriately measured using a Poisson or negative binomial distribution. In any case, the authors should indicate that their data satisfy the assumptions of whichever model they choose to use. It's my understanding that the Tukey HSD test is post-hoc so you should report the primary test (e.g. ANOVA) and then use Tukey HSD to determine differences. At times Tukey HSD is reported alone.

One of the authors is a well-known taxonomist with expertise in the region. However, there remain some issues with the treatment of the taxonomic component of the study. 1) Morphospecies are listed in the Appendix without even a family level identification. 2) Taxon authorities are not listed next to species that are identified. 3) The authors cite only two papers (Ayala 1999, Michener 2000) for the taxonomy, however the taxon concepts for the species do not and cannot come from these studies alone. I realize that taxonomists often know species by sight, but that doesn't mean the taxon concepts don't rely on the previous work of others. These should be cited in the same way you would cite an R package (p.s. you should cite your R packages, too!)

Validity of the findings

I would like to see a more careful analysis of the data. I don't think all the conclusions are necessarily supported by robust statistics.

Additional comments

This seems like a good effort to me, but the limited number of sample sites need to be dealt with carefully.

·

Basic reporting

The overall manuscript is of high quality and self-contained.

The manuscript is very well-written in an English language that is clear and unambiguous. I did, however, notice a few minor linguistic mistakes, which I indicated in the attached PDF document.

The introduction is well-written and demonstrates how this research contributes to existing knowledge gaps. However, to my opinion, some background on the pollination of coffee is missing. This should not go into too much detail, as the actual pollination of coffee flowers is not studied in this research, yet some background would be useful in order to motivate why it is important to study bee diversity in Arabica coffee farms. Since the importance of pollinators and their diversity is different for Coffea arabica compared to Coffea canephora, I would also advice not to wait until the Materials and methods section to mention that this study concerns Coffea arabica.
The provided background information is supported by sufficient and relevant literature references.

The manuscript is structured in a professional way.

Figures and tables are of high quality, relevant and appropriately described. A few minor remarks:
- Legend of Figure 4: the description of graphs b and c should be switched.
- Legend of Figure 6: I think ‘(B) Bee richness’ in the legend should be replaced by ‘(S) Bee richness’?
- Table 2: It is not explained in the legend why certain estimate-values are presented in bold.

Raw data have been made available in appendix.

Experimental design

This study presents original primary research that fits within the aims and scope of the journal.

The research questions are well defined, relevant and meaningful.

The methods are described in a detailed way. I do, however, have some minor remarks:

- A map showing the location of the three studied coffee farms could be a nice addition as it would provide information about the distance between these studied farms, their size, and the area and position of the surrounding secondary forests and other land-use types (e.g., villages or primary forest). This is some valuable information that I think is lacking in the manuscript.

- Perhaps some short motivation could be presented as to why the bees were grouped into honeybees, native bees, and stingless bees. What is the main ecological difference between these three groups of bees?

- The reference to the different plot types (coffee, early, late) throughout the text and tables/figures is not always consistent. Sometimes this is referred to as ‘forest type’, ‘plot type’, ‘vegetation type’, or ‘landscape type’. To avoid confusion, I would recommend to stick to one term only.

Statistical analysis methods are carefully selected and the choices made are motivated.

Validity of the findings

No comment.

Additional comments

Dear authors, to my opinion, you performed an interesting and relevant study in a well-organized and high-quality way, and you wrote this manuscript in the same way. It was a pleasure to read this well-written manuscript.

---

## Round 0.2 · accepted · Accept

Thank you for addressing the reviewer comments. Congratulations!